# Emotional discourse analysis of COVID-19 patients and their mental health: A text mining study

Yu Deng[1], Minjun Park[2], Juanjuan Chen[3], Jixue Yang[4], Luxue Xie[4], Huimin Li[4], Li Wang[5], Yaokai Chen[6]*

1 College of Language Intelligence, Sichuan International Studies University, Chongqing, China, 2 Chinese Language and Literature, Duksung Women's University, Seoul, Republic of Korea, 3 Institute of Educational Planning and Assessment, Sichuan International Studies University, Chongqing, China, 4 School of English, Sichuan International Studies University, Chongqing, China, 5 Science and Education Department, Chongqing Public Health Medical Center, Chongqing, China, 6 Division of Infectious Diseases, Chongqing Public Health Medical Center, Chongqing, China

☯ These authors contributed equally to this work.
* yaokaichen@hotmail.com

**Data Availability Statement:** The minimal dataset is available within the paper. Additional data is available in figshare: https://doi.org/10.6084/m9.figshare.20522775.

## Abstract

COVID-19 has caused negative emotional responses in patients, with significant mental health consequences for the infected population. The need for an in-depth analysis of the emotional state of COVID-19 patients is imperative. This study employed semi-structured interviews and the text mining method to investigate features in lived experience narratives of COVID-19 patients and healthy controls with respect to five basic emotions. The aim was to identify differences in emotional status between the two matched groups of participants. The results indicate generally higher complexity and more expressive emotional language in healthy controls than in COVID-19 patients. Specifically, narratives of fear, happiness, and sadness by COVID-19 patients were significantly shorter as compared to healthy controls. Regarding lexical features, COVID-19 patients used more emotional words, in particular words of fear, disgust, and happiness, as opposed to those used by healthy controls. Emotional disorder symptoms of COVID-19 patients at the lexical level tended to focus on the emotions of fear and disgust. They narrated more in relation to self or family while healthy controls mainly talked about others. Our automatic emotional discourse analysis potentially distinguishes clinical status of COVID-19 patients versus healthy controls, and can thus be used to predict mental health disorder symptoms in COVID-19 patients.

## Introduction

A deadly global pandemic is expected to be associated with various emotions in different people, and might cause negative emotional responses, with serious consequences for mental health [1, 2]. The emergence of the COVID-19 (coronavirus disease 2019) has raised significant questions about the mental health burden of the population relating to social restrictions, lockdowns, school and business closures, loss of livelihoods, decreases in economic activity,

**Funding:** This research was funded by Humanities and Social Sciences Research Project of Chongqing Education Commission in the form of a grant (21SKGH143) awarded to YD. This study was also funded by Foundation of First-class Discipline of Foreign Languages & Literature, Chongqing in the form of a grant (SISUWYJY202104) awarded to YD. This study was also funded by Teaching reform project of Sichuan International Studies University in the form of a grant (JY2296294) awarded to YD. This study's APC was funded by Chongqing Talent Cultivation Program in the form of a grant (cstc2021ycjh-bgzxm0275) awarded to YC. The funders had no role in study design, data collection and analysis, decision to publish, or preparation of the manuscript.

**Competing interests:** The authors have declared that no competing interests exist.

and shifting priorities of governments in their attempt to control COVID-19. Notably, COVID-19 may result in significant physical and mental consequences among those who become infected. While the physical treatment modalities for COVID-19 have received extensive attention, the need for exploration of the mental health impact of SARS-CoV-2 infection on patients has never been more urgent [3].

After the outbreak of the current COVID-19 pandemic, a growing body of survey studies concerning the emotional state and mental health of COVID-19 patients have shown that patients afflicted by COVID-19 experience post-traumatic stress disorder, boredom, loneliness, anger, insomnia, anxiety, depression, and distress [4–10]. What the preceding studies have in common is that they show that the acquisition of COVID-19 leads to an increase in negative emotions and a decrease in positive emotions for infected patients.

While a considerable number of surveys have been conducted to investigate the mental health problems of COVID-19 patients, it seems that the lived experience narrative discourse concerning the dynamic psychological and emotional experiences of COVID-19 patients are currently underestimated (e.g., [11–20]). Notably, Missel et al. [14] used telephone interviews to explore the lived experiences of 15 individuals infected with COVID-19 in Denmark. Qualitative analysis demonstrated that being diagnosed with COVID-19 was seen as a threat to patients' existence and to their bodily perception, as well as to social relationships. COVID-19 patients experience unpleasant emotions given the negative social interactions that COVID-19 causes. Venturas et al. [19] conducted inductive, in-depth interviews with 11 hospitalized COVID-19 patients in Spain, and their phenomenological qualitative results indicated that although patients were uncertain about the coronavirus and felt frustrated about hospital isolation, they positively adapted to the situation and felt confident and safe in the care of medical staff. Efficient high-tech communication with relatives had somehow alleviated their negative emotions. Son et al. [16] interviewed 16 discharged COVID-19 patients in South Korea, and the phenomenological interpretation of their patient narratives showed that COVID-19 patients experienced social stigma and feelings of guilt due to negative attitudes from society and social media. However, the patients developed positive emotions through the social support of families and friends. In the context of China, Sun et al. [17] conducted semi-structured interviews with 16 COVID-19 patients from Henan, China. Patients talked about their feelings during hospitalization and isolation. Results revealed diachronic changes to their emotional experiences. Specifically, patients' emotional responses toward the infection, which included fear, denial, and stigma during the acute phase of infection, gradually changed into positive emotions (i.e., acceptance, trust) in the later recovery phase. Through a qualitative study of the lived experience narratives of 16 hospitalized COVID-19 survivors from Nanning, China, Wu et al. [20] identified anxiety, trauma, and self-stigmatization among COVID-19 patients. Their negative emotional responses were strongly associated with social interactions, suggesting that it is urgent to minimize negative social impacts on COVID-19 infected individuals. Li et al. [13] used a phenomenological approach in a qualitative study of 13 hospitalized COVID-19 patients in Wuhan. Inductive thematic analysis of patient narrative discourse showed that COVID-19 patients experienced negative emotions such as confusion, uncertainty, worry, and guilt. The negative emotions mainly concerned anxiety regarding social discrimination and poor financial security. Their positive emotions were aligned with expectations of life after recovery, abundant social support from healthcare workers and family, as well as the strength of the government. Deng et al. [11] conducted semi-structural interviews with 6 re-positive patients from Wuhan and Chongqing to elicit their narrative discourse of recurrent COVID-19 infections. Their results revealed that SARS-CoV-2 nucleic test re-positive patients demonstrated emotional problems such as anxiety, depression, irritation, stress, mistrust, insomnia, suicidal tendency, grief, panic, and worry. Social isolation, lack of emotional support, and

negative news in the mass media were the main risk factors leading to their mental health problems.

Clearly, the narrative discourse of the emotional experience of COVID-19 acquisition encompasses rich personal information in linguistic form [21]. Published literature concerning the lived experiences of COVID-19 patients reveals that emotional discourse is an important marker providing information about patients' internal mental representations of the COVID-19 pandemic. To date, the emotional discourse analysis of patients' lived experiences of COVID-19 extensively focuses on the qualitative approach [22], whereas the quantitative approach is currently underrepresented. Recent studies have used natural language processing (NLP) techniques to extract real-world text data concerning COVID-19 from online resources such as Twitter, Facebook, and Reddit, in order to provide automated population-level health surveillance [23–26]. For instance, Crocamo et al. [23] conducted sentiment analysis of 3,308,476 English tweets concerning COVID-19 discussions between January 19 and March 3, 2020, by computing a polarity compound score and using a transformer-based model. Their results demonstrated an increasing trend towards negative sentiment as the COVID-19 pandemic proceeded. Low et al. [24] used NLP techniques to detect changes in mental health support groups (e.g., Schizophrenia, Suicide Watch, Depression) and non-mental health groups (e.g., Personal Finance, Conspiracy Theory) at the outbreak of COVID-19. Drawing on the Reddit data of 826,961 users by using unsupervised methods such as topic modeling and clustering, Low et al. found that support groups related to attention-deficit/hyperactivity disorder, eating disorders, and anxiety showed the most negative semantic changes during COVID-19. Health anxiety was found to be a salient theme across Reddit through machine learning analyses. Patel et al. [25] used data mining methods to compute 739,434 COVID-19 related posts by 53,134 users from online health forums such as Health Boards and Inspire & Health Unlocked from 1 January 2020 to 31 May 2020. Their results showed that the global lockdown policy had led to an increase of discussion related to COVID-19-related disorders, and 25% of the COVID-19-related titles mentioned an associated physical or mental health comorbidity. Viviani et al. [26] conducted word identity and sentiment analysis of vulnerability to psychological distress in the COVID-19 context by gathering texts related to COVID-19 on the Twitter microblogging platform between August and December 2020. The automatic analysis indicated that the social distancing scenario was associated with 'annoyance' and 'protest', while the vaccine scenario was linked to 'killing' and 'murder'. Regarding anxiety, the target scenarios of social distancing and symptoms & hospitalization were closely related to risk, worry, and fear. As for vulnerability scores in the word categories, the lexicon identifiers regarding greater psychological vulnerability fell into the 'anger' and 'risk' categories, in particular for social distancing and vaccines & vaccinations. There was a large increase in vulnerability identifiers in the 'death' category shared by the 'symptoms & hospitalization' scenario.

Text mining based on real-world data is a sensitive method to reveal emotional and mental health burden during COVID-19, given that it provides an effective marker to identify the mental status of vulnerable groups and alarming themes during the ongoing pandemic [24]. It is noteworthy that the automatic analysis of COVID-19 patients' texts is somehow overlooked in the literature. Along this line of enquiry, the present study combined in-depth semi-structured interviews with text mining methods to investigate features in lived experience narratives regarding COVID-19 in persons with COVID-19 and in healthy controls related to five basic emotions. Our aim was to identify differences in emotionality between COVID-19 patients and healthy controls based on narrative discourse, in which the subjects describe their past emotional experiences of COVID-19. We hypothesize that mental health variables can be inferred across different emotions through linguistic markers in the narrative discourse [27, 28].

## Materials and methods

### Study design

The present study combined qualitative and quantitative approaches in data collection and text analysis. The narrative data emerged from a larger cross-sectional project, i.e., "Lived-Experience Narratives and Mental Health of People in China during the COVID-19 Pandemic" [11–12, 22, 29]. The project used semi-structured interviews to collect narrative discourses concerning lived experiences of COVID-19 in China, from the period from 30 June 2020 to 31 January 2021. The project teams recruited participants by convenience sampling. COVID-19 patients were randomly recruited from Chongqing Public Health Medical Center, China. The inclusion criteria for this study included those COVID-19 patients who had been hospitalized for at least 14 days and who were required to quarantine at home or at a hotel for a further 14 days after being discharged from hospital [12]. The healthy controls, who had experienced the first wave of the COVID-19 pandemic but were not infected by SARS-CoV-2, were randomly selected from citizens of Wuhan, China. Due to the COVID-19 prevention policies, telephone interviews were employed to ensure the personal safety of the interviewer and participants [20]. Participants' narrative discourses of lived experiences involved multiple dimensions of their emotional and mental health status. The term "lived experiences" designates the phenomenological tradition regarding experiences of the everyday life world. Such lived experiences are prereflective and less available to our awareness [14]. Thus, the lived experience narratives are capable of capturing areas of participants' mental health status that cannot be explored by the survey method [12, 14]

The lived experience narrative discourse was then annotated qualitatively according to the five primary emotions [21] and the 21 emotional subcategories in the Chinese Affective Lexicon Ontology [30]. In the discourse that follows, the generic features of the emotional narratives (i.e., mean word-length, words per sentence, sentences per narrative, and words per narrative), frequency of emotional words, and significant word identity features between the two matched groups were computed quantitatively by text mining methods. Based on text analysis of emotional narratives, differences between persons infected with COVID-19 and healthy controls were measured with respect to each of their mental health statuses during the COVID-19 pandemic.

### Data collection

We collected lived experience narratives from 58 participants (34 persons infected with COVID-19 and 24 healthy controls). Demographic and clinical information for this cohort is provided in Table 1. Written informed consent was obtained from all individual participants before their interview. Ethics committee approval was received for this study from the ethics committee of the Ethics Committee of Chongqing Public Health Medical Center (ID: 2020-

**Table 1. Participant information.**

| Variables | Full sample (N = 58) | COVID-19 patients (N = 34) | Healthy Controls (N = 24) |
|---|---|---|---|
| Gender: male | 29 | 18 | 11 |
| Gender: female | 29 | 16 | 13 |
| Marital status: married | 34 | 24 | 10 |
| Marital status: unmarried | 22 | 8 | 14 |
| Marital status: divorced | 2 | 2 | N/A |
| Mean age (S.D.) | 34.42 (10.70) | 38.82 (9.95) | 27.91 (8.28) |
| Mean time in hospital (S.D.) | | 17.29 days (13.67) | N/A |

048-02-KY). All procedures involving human participants were in accordance with the ethical standards of the institutional and national research committees and with the Helsinki declaration and its later amendments, or with comparable ethical standards.

Participants attended the in-depth semi-structural interview by telephone, as paid volunteers. They were asked to narrate their personal experiences and feelings concerning COVID-19 during the interview. The interview domain concerned multiple dimensions of their lived experience during COVID-19, such as work and daily life, social isolation, family support, social relationships, attitude towards death, comments on medical service, and national policy [11, 12, 22, 29]. Patient interview questions included "How were you infected with COVID-19", "What did you do before, during, and after hospitalization?", "How were your work and daily life impacted due to the infection?", "How did you feel when you were isolated in the hospital?", "What gave you comfort when you were isolated?", "What changes in perceptions of the world do you have?", "How did you deal with social relationship after discharge", "What was your attitude toward death after the contraction of COVID-19?", "What was the most unforgettable event you experienced during the pandemic?", "What is your opinion about the role our country and medical staff play in combating COVID-19?"[11, 12, 22]. Healthy controls' interview questions included the following: "What did you do at the outbreak of COVID-19?", "How was your work and daily life impacted by the pandemic?", "What gave you comfort when you felt frustrated?", "What changes in perceptions of the world did you have?", "How did you feel in witnessing death every day?", "How did you feel when the city was locked down?", "What was the most unforgettable event you experienced during the pandemic?", "What was your attitude toward death during COVID-19?", "What is your opinion about the role our country and medical staff play in combating COVID-19?" [29].

The telephone interviews were conducted during two periods, namely from June 2020 to August 2020 and from October 2020 to January 2021, by two researchers specializing in psychology and psycholinguistics. The interviews were arranged during summer and autumn-winter as dictated by participants' requirement and their pandemic situation. Each interview lasted approximately 40–60 minutes, and was audio-recorded. The recorded audio files were transcribed verbatim by the interviewers to allow for text analysis.

## Data processing

All the recorded interview files were transcribed verbatim. The raw COVID-19 interview corpus consisted of 326,202 words. The raw data encompassed conversations between the interviewer and interviewees. Given that the aim of this study was to investigate the narrative discourse of participants, we used a Python script to exclude interviewer questions and utterances, leaving behind only those narratives of the interviewees. Furthermore, typographical errors and inconsistencies occurring among different transcribers were corrected. We obtained a corpora sample size of 167,795 word tokens for the 34 COVID-19 patients, and a sample size of 121,372 word tokens for the 24 healthy controls. Emotional narrative tagging and text preprocessing were conducted for the purpose of measuring sentiment distribution and linguistic features between the two matched groups.

## Emotional narrative tagging

Based on the text of the interviewees discourses, three researchers conducted sentiment classification and manual annotation of the emotional narratives [22]. In distinguishing universal emotions, Ekman [31] classified the six basic emotions as anger, disgust, fear, happiness, sadness, and surprise. Hong et al. [21] optimized Ekman's classification as five types—happy, angry, sad, fear, and disgust. Based on Ekman's six basic emotions, Xu et al. [30] established

**Table 2. Emotional categorization and labels [22].**

| No | Subcategorization [30] | Primary Emotions [21] | Polarity |
|---|---|---|---|
| 1 | Joy (PA) | HAPPY | Positive |
| 2 | Comfort (PE) | HAPPY | Positive |
| 3 | Respect (PD) | HAPPY | Positive |
| 4 | Praise (PH) | HAPPY | Positive |
| 5 | Trust (PG) | HAPPY | Positive |
| 6 | Like (PB) | HAPPY | Positive |
| 7 | Wish (PK) | HAPPY | Positive |
| 8 | Angry (NA) | ANGRY | Negative |
| 9 | Upset (NB) | SAD | Negative |
| 10 | Disappointed (NJ) | SAD | Negative |
| 11 | Guilty (NH) | SAD | Negative |
| 12 | Grief (PF) | SAD | Negative |
| 13 | Panic (NI) | FEAR | Negative |
| 14 | Dread (NC) | FEAR | Negative |
| 15 | Shame (NG) | FEAR | Negative |
| 16 | Depressed (NE) | FEAR | Negative |
| 17 | Hate (ND) | DISGUST | Negative |
| 18 | Criticize (NN) | DISGUST | Negative |
| 19 | Envious (NK) | DISGUST | Negative |
| 20 | Suspect (NL) | FEAR | Negative |
| 21 | Surprise (PC) | FEAR/HAPPY | Negative/Positive |

Note: The initials in brackets are the original ID tags in Xu et al.,'s [30] emotion dictionary. P stands for positive and N for negative.

the Chinese Affective Lexicon Ontology, which includes seven basic traditional Chinese emotions called *qi qing*七情 (i.e., *fear* 惧, *anger* 怒, *joy* 乐, *sadness* 哀, *disgust* 恶, *surprise* 惊, *and good* 好), and 21 emotional subcategories over 27,466 lexical entries. The 21 emotional subcategories have largely extended the range of emotional experience analysis. To minimize the basic emotions and enlarge the subcategories of emotions, the present study combined the five primary emotions of Hong et al. [21] and the 21 emotional subcategories of Xu et al. [30] in emotional narrative tagging (Table 2). We first manually coded the emotional narratives, which referred to each discourse unit that encompassed congruent topics or emotions within the interview texts [21, 22]. Three researchers coded the discourse unit of emotional narratives together. Inconsistencies and disagreements were settled by a fourth researcher. Five translated exemplars of emotional narratives are shown below.

> ++HAPPY (PH/PD) ++: *The medical staffs were so great. I found that they were young girls when they took off the protective equipment. They persisted in helping the COVID-19 patients and were so nice to patients. They are so great and selfless in our eyes. (Patient 2)*

> ++SAD (NB) ++: *On the way to the isolation hospital, I cried all the way and felt extremely upset. (Patient 3)*

> ++ANGRY (NA) ++: *The internet is convenient. My personal information was exposed on the net. This made me irritated. I showed them the official document of negative test result for COVID-19. However, people in our community still refused to let me in. I felt extremely angry at that time. They showed discrimination to persons who had contracted COVID-19. (Patient 27)*

*++FEAR (NC) ++: I felt very fearful, extremely fearful. My heart beat fiercely. I felt that my heart beat the fastest in my life. I was very nervous and scared. I felt at a loss. My heat beat even 240 times per minute at that time. (Patient 6)*

*++DISGUST (ND) ++: I vomited when I smelled the flavor of the medicine. I kept taking the medicine, but vomited all the time. The doctor prescribed some anti-omitting medicine for me, but it did not work. Every time I saw the medicine, I felt disgusted and uncomfortable. (Patient 7)*

*As these excerpts show, emotional narratives were classified in terms of five basic emotions and 21 emotional subcategories, by using five primary emotional tags plus subcategories, namely ++HAPPY (x)++, ++SAD(x)++, ++ANGRY(x)++, ++FEAR(x)++, and ++DISGUST(x)++. For narratives lacking emotion, the tag ++EMPTY++ was used.*

**Text preprocessing.** The structured narrative corpus was then subjected to grammatical processing. First, individual discourses were divided into sentences and paragraphs based on punctuation marks. Word tokens, which are fundamental linguistic units, were obtained through word segmentation of individual sentences. It is worth noting that Chinese word segmentation differs from English tokenization due to the absence of a distinct boundary between individual Chinese words. Hence, the Chinese word segmentation task was performed using the LTP toolkit of Harbin Institute of Technology [32], and incorrectly segmented parts of sentences were corrected manually. The segmented corpus was used for the analysis of generic features and word identity features of emotional narratives between the two matched groups. Second, regarding the analysis of sentiment polarity and 5/21 types of emotional words, irrelevant functional words such as prepositions, auxiliary verbs, and conjunctions were removed from the corpus in order to maximize the effect of sentiment analysis. Functional words are syntactically meaningful, but undermine the macroscopic semantic interpretation of the emotional words. By removing the preceding 'noise' elements, we obtained a sample size of 141,878 words (functional words excluded) for the patient group, and a sample size of 106, 499 words (functional words excluded) for the healthy controls.

## Measures

Based on the preprocessed corpus labeled with emotional tags, quantitative analysis of the emotional narrative followed. The first step involved automated analysis of generic features of the emotional narratives of COVID-19 patients and healthy controls, including mean word-length, words per sentence, sentences per narrative, and words per narrative. In the second step, sentiment analysis was implemented by calculating the frequency of emotional words. In doing so, differences regarding polarity and intensity of emotions were exposed between patients and controls. Lastly, significant features of word identity across the emotional narratives were compared between the two matched groups.

**Analysis of generic features of emotional narratives.** In the present study, generic features of emotional narratives, including Type-Token ratio, mean word-length, average number of words per sentence, number of sentences per narrative, and number of words per narrative were measured among patients and healthy controls. The Type-Token ratio shows the number of unique words divided by the total number of words in the emotional narratives corpus. A lower ratio indicates more use of the same words in the text. Mean word-length refers to the average number of Chinese characters per word. Longer words are usually more complex and indicates better use of language. Regarding average number of words per sentence, longer

sentences tend to be syntactically more complex and express more complex ideas. As for number of sentences per narrative, more sentences in a specific emotional narrative of similar length indicate the use of shorter, less complex sentences. Total number of words per narrative specifies word frequency in a specific emotional narrative [21].

The emotional narratives corpus was loaded into Python for generic feature analysis, drawing on the 'CorpusReader' class in NLTK (Natural Language Toolkit)—an open source Python library for natural language processing [33]. This package made file opening and stream control robust, and lent support to large-scale text retrieving and operating of the current corpus. As CorpusReader class in NLTK provides various text processing measurements (e.g., paras(), sents(), words()), it was used to automatically measure grammatical categories, such as word, sentence, and paragraph, in specific narratives of patients and controls (with fileids, categories argument). In this regard, average X per Y = len(X) / len(Y) was computed with various subfunctions of CorpusReader classes including reader.words(), reader.sents(), reader.paras() in conditions such as reader.words (categories = 'patient').

**Analysis of frequency of emotional words.**   To compute sentiment statistics, we used the package called CategorizedCorpusReader.words(categories = 'group') in NLTK to calculate frequency of emotional words from specific group of interest (e.g., patient or control, positive or negative, HAPPY or SAD). First, words with no substantive meaning (functional words, punctuations) were filtered out, based on the stopwords list of Harbin Institute of Technology [32]. Next, frequency of emotional polarity and 5/21 categories of emotional words were extracted according to the Chinese Affective Lexicon ontology [30], by utilizing 'probability.ConditionalFreqDist' class in NLTK. By doing so, a conditional frequency distribution table was generated. ConditionalFreqDist (CFD) is essentially word frequency data stored in key:value pairs, where 'key' is the user-defined condition and 'value' is word frequency distribution (a 'FreqDist' class) of a group satisfying the condition. CFD is able to create individual 'FreqDist' objects with specific conditions of various sentiments (i.e., polarity, primary, or subcategorized emotions) and source of observation (i.e., patient or control).

Utilizing these tools, we conducted sentiment analysis by selecting the frequency of emotional vocabulary for each group according to different conditions. For instance, the occurrence of the vocabulary group of the primary emotion 'DISGUST' among healthy controls was retrieved with the query (CFD['control']['disgust']). Other emotional types were measured in a similar way (e.g., CFD ['patient']['NC'] calculated the frequency of fear related words among the patient group). Apart from searching for individual observation values, we used the plot function in 'FreqDist' and the Python library, 'Matplotlob', to visualize the frequency of different types of emotional words by groups.

**Analysis of significant word identity features of emotional narratives.**   Significant word identity features, such as the probabilistic tendency of specific words, can mark mental status of COVID-19 patients as compared to healthy controls. In this study, the Log-likelihood ratio (LL) was adopted to measure the difference in word identity between the two sub-corpora. Table 3 shows the rationales of the comparison and mathematical formula [34–36] that were implemented via 'AntConc' software. Table 4 shows a comparison of word identity in SAD narratives between the two matched groups. Looking at 'not' and 'no' in Table 4, the high LL

**Table 3. Rationale of Log-likelihood ratio (LL) for word identity comparison [34–36].**

|  | Patient Narratives (i*) | Control Narratives (i*) | Total |
|---|---|---|---|
| **Frequency of target word (O*)** | a | b | a+b |
| **Frequency of other words** | c-a | d-b | c+d-a-b |
| **Total (*N)** | c | d | c+d |

**Table 4. Most distinctive words of SAD narratives between the two matched groups.**

| Word Feature | Log-likelihood | Patient (N = 34) | | Control (N = 24) | | Overuse/Underuse |
|---|---|---|---|---|---|---|
| | | Frequency | Relative Frequency | Frequency | Relative Frequency | |
| *then* | 52.53 | 178 | 0.63% | 226 | 1.30% | - |
| *not* | 48.14 | 609 | 2.16% | 224 | 1.29% | + |
| *Wuhan* | 46.30 | 11 | 0.04% | 48 | 0.28% | - |
| *anyway* | 43.98 | 70 | 0.25% | 4 | 0.02% | + |
| *possibly* | 42.66 | 79 | 0.28% | 123 | 0.71% | - |
| *isolation* | 41.81 | 80 | 0.28% | 7 | 0.04% | + |
| *no* | 39.47 | 160 | 0.57% | 34 | 0.20% | + |
| *for sure* | 32.94 | 61 | 0.22% | 5 | 0.03% | + |
| *days* | 29.95 | 130 | 0.46% | 29 | 0.17% | + |
| *want* | 28.81 | 192 | 0.68% | 55 | 0.32% | + |
| *life* | 27.14 | 9 | 0.03% | 32 | 0.18% | - |
| *sister* | 25.99 | 27 | 0.10% | 0 | 0.00% | + |
| *world* | 25.98 | 1 | 0.00% | 17 | 0.10% | - |
| *patient* | 25.64 | 7 | 0.02% | 28 | 0.16% | - |
| *worry* | 25.57 | 34 | 0.12% | 1 | 0.01% | + |
| *infectious* | 25.02 | 26 | 0.09% | 0 | 0.00% | + |

Note: Log-likelihood values of 15.13 or higher are significant (d.f. = 1, p<0.0001)

value suggests that patients narrated more negative experiences or attitudes compared to healthy controls. The same is true of words such as '*isolation*', '*days*', '*want*', '*worry*', and '*infectious*', which concerned patients' personal experiences of infection, quarantine, and treatment. In contrast, words such as '*Wuhan*', '*life*', '*world*', and '*patient*' were used more frequently in healthy controls compared to the patient group, suggesting that healthy controls were more concerned with external circumstances such as the community and the country rather than themselves.

$$-2 \ln \lambda = 2 \sum_i O_i \ln \left( \frac{O_i}{E_i} \right), \quad where\ E_i = \frac{N_i \sum_i O_i}{\sum_i N_i}$$

## Results

### Generic features of emotional narratives

The statistical results of generic features in the emotional narratives between COVID-19 patients and healthy controls are shown in Table 5. The type-token ratio (TTR) of vocabulary was used to measure lexical diversity [37]. Mean word-length was also an essential factor in detecting psychological processes of speakers in communication. With regard to mean word-length in sentences, emotionally traumatized patients tended to use longer sentences and irrelevant words for expression of a simple idea. In the literature, a smaller number of sentences in patient narratives and smaller number of words per narrative suggest possible frequent changes of topic caused by an unstable emotional state [21].

In our dataset, COVID-19 patients and healthy controls did not differ in the type-token ratio (t = 0.6, df = 56, $p$ = 0.551). There was a slight difference in mean word-length, which was under the marginal significance level (t = 1.99, df = 55.1, $p$ = 0.051). Notably, COVID-19 patients employed significantly more words per sentence (t = -5.75, df = 52.6, $p$<0.001), had

**Table 5. Average value of generic features for COVID-19 patients and healthy controls.**

| Features | Patient (N = 34) | Control (N = 24) | Welch's t | p |
|---|---|---|---|---|
| Type/token ratio | 0.199 | 0.191 | 0.60 | 0.551 (df = 56.0) |
| Normalized | 0.307 | 0.342 | 0.60 | 0.551 (df = 56.0) |
| Mean word-length | 1.49 | 1.51 | 1.99 | 0.051 (df = 55.1) |
| Normalized | 0.419 | 0.532 | 1.99 | 0.051 (df = 55.1) |
| Words/sentence | 23.7 | 19.5 | -5.75 | < .001 (df = 52.6) |
| Normalized | 0.510 | 0.247 | -5.75 | < .001 (df = 52.6) |
| Sentences/narrative | 2.41 | 4.11 | 6.12 | < .001 (df = 30.3) |
| Normalized | 0.18 | 0.50 | 6.12 | < .001 (df = 30.3) |
| Words/narrative | 48.4 | 70.3 | 3.62 | < .001 (df = 37.4) |
| Normalized | 0.27 | 0.49 | 3.62 | < .001 (df = 37.4) |

Note: After min-max normalization, two-tailed, independent samples t-test

significantly fewer sentences per narrative (t = 6.12, df = 30.3, $p<0.001$), and used significantly fewer words per narrative (t = 3.62, df = 37.4, $p<0.001$).

Detailed information for the number of words per narrative by the five basic emotions between COVID-19 patients and healthy controls is shown in Table 6 and Fig 1. When comparing overall narrative length by emotion, COVID-19 patient narratives of fear, happiness, and sadness were significantly shorter as compared to healthy controls. It is noteworthy that although narratives of anger and disgust were not statistically significant, as a whole, healthy controls were likely to express different emotions through a higher number of words as compared with patients, except for narratives of anger.

## Frequency of emotional words

We focused on examining frequency distribution of emotional words related to sentiment polarity and the 5/21 emotional classification. The frequency distribution of emotional words between the two matched groups is shown in Figs 2–4. Regarding the overall distribution of the emotionality, COVID-19 patients generally used more emotional words than healthy controls, regardless of using negative or positive words (see Fig 2). For emotional words across the five emotions of happiness, fear, disgust, sadness, and anger, COVID-19 patients generally used more words of fear, disgust, and happiness, as compared to healthy controls (see Fig 3). To calculate the frequency of emotional words precisely, we computed the frequency of emotional words in terms of Xu et al.'s [30] 21-type classification. The results are shown in Fig 4. It is clear that COVID-19 patients used more emotional words of NC (dread), NE (depressed), NN (criticize), PA (joy), PG (trust), and PB (like) than the healthy controls. According to Table 2, emotional words of NC and NE fall into the emotion of fear, NN belongs to the emotion of disgust, and PA, PG, and PB are classified into the emotion of happiness. The result of

**Table 6. The average number of words per narrative across five emotions.**

| Narrative types | Patient (N = 34) | Control (N = 24) | Welch's t | p |
|---|---|---|---|---|
| ANGRY | 91.8 | 71.8 | 0.990 | 0.327 |
| DISGUST | 73.6 | 90.8 | -1.51 | 0.133 |
| FEAR | 72 | 102 | -8.80 | < .001 |
| HAPPY | 62.8 | 80.6 | -5.71 | < .001 |
| SAD | 62.9 | 96.5 | -5.56 | < .001 |

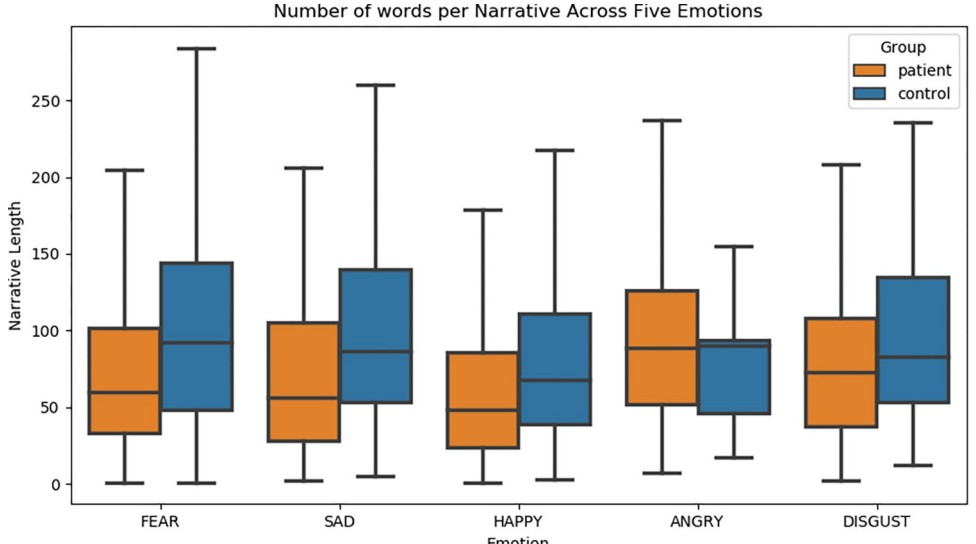

**Fig 1. The average number of words per narrative across five emotions.**

the 21-type classification of emotional words was consistent with the 5-type classification, as seen in Figs 3 and 4. Taking the 5-type and 21-type classifications together, mental health issues of COVID-19 patients seem to have been concentrated on the emotions of fear and disgust. Given that COVID-19 patients had experienced the processes of infection, isolation, and recovery, their emotions fluctuated more dramatically when compared to healthy controls, as depicted in Fig 4. Although patients initially experienced frustrated emotional phases at the

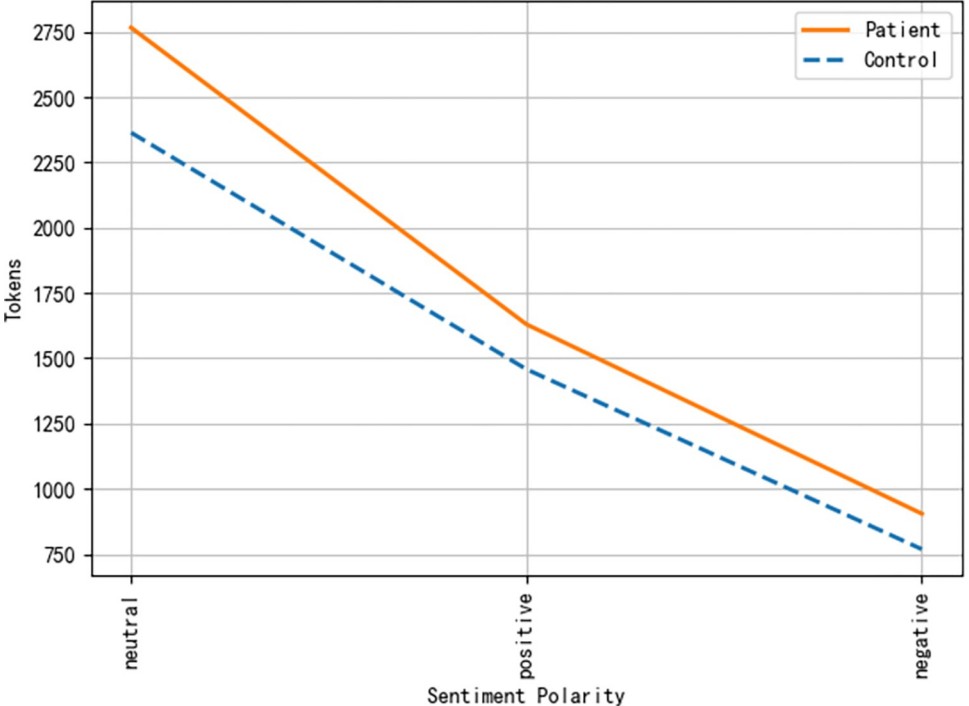

**Fig 2. Frequency of positive and negative words between COVID-19 patients and healthy controls.**

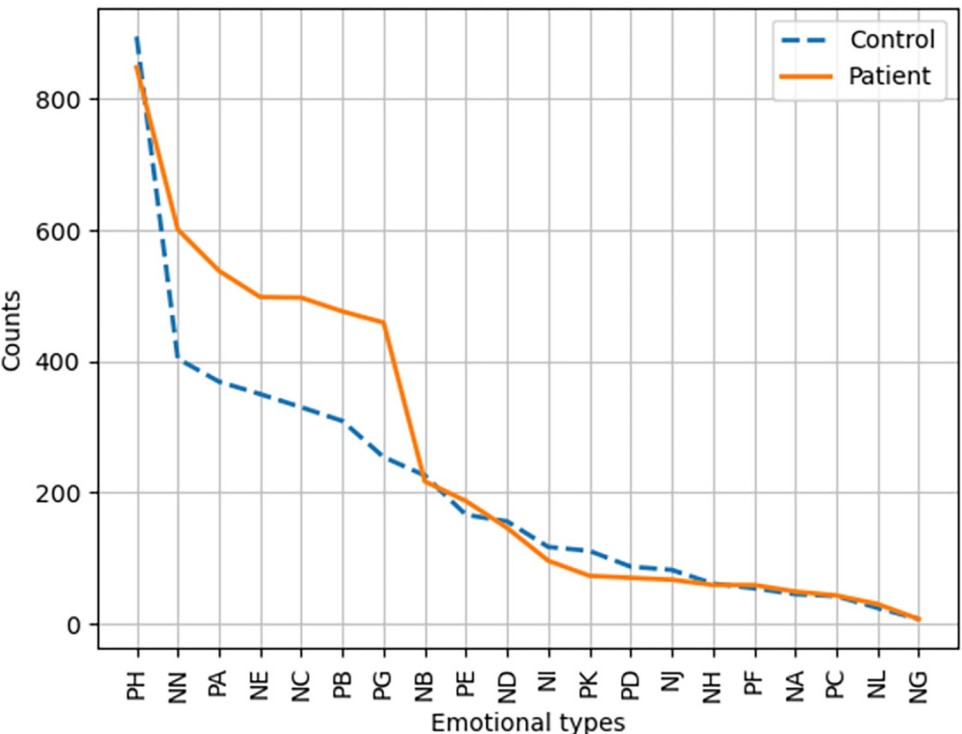

**Fig 4. Frequency of the 21 types of emotional words between COVID-19 patients and healthy controls.** The 21-type ID tags: Joy (PA), Comfort (PE), Respect (PD), Praise (PH), Trust (PG), Like (PB), Wish (PK), Angry (NA), Upset (NB), Disappointed (NJ), Guilty (NH), Grief (PF), Panic (NI), Dread (NC), Shame (NG), Depressed (NE), Hate (ND), Criticize (NN), Envious (NK), Suspect (NL), Surprise (PC).

time of confirmed infection, their final recovery made COVID-19 patients happy and optimistic, and they trusted the government and medical staff in a positive emotional manner.

## Significant word identity features of emotional narratives

To demonstrate the significant features of word identity between COVID-19 patients and healthy controls, we computed the word identity features across narratives of five basic emotions by means of the Log-likelihood difference test. Results are shown in Table 7.

Looking at Table 7, we observed that COVID-19 patients were more likely to use the first person pronouns "I" and "we" to describe their five emotions, suggesting that patient emotional narratives were more about self or family surviving the pandemic, while healthy controls mainly talked about others (e.g., some people, Wuhan citizens, Red Cross, medical staff). Patients' negative emotions (anger, disgust, fear, sadness) were mainly concerned with the topics of infection and reinfection, disease symptoms, nucleic acid testing, isolation in hospital and hotel, worry about infecting family and friends, lack of social support, and stigma. In contrast, negative emotions for healthy controls were mainly about the lockdown of the city, the community, medical staff, volunteers, patients, the prevailing living conditions, and the protective measures. With respect to the positive emotion (i.e., happiness), COVID-19 patients talked more about their journey of recovery, namely about how they were cured by medical staff and how they dealt with pressure successfully. The positive emotions of healthy controls were mainly concerned with how the city, the community, and the hospitals operated in an orderly way by collective community efforts.

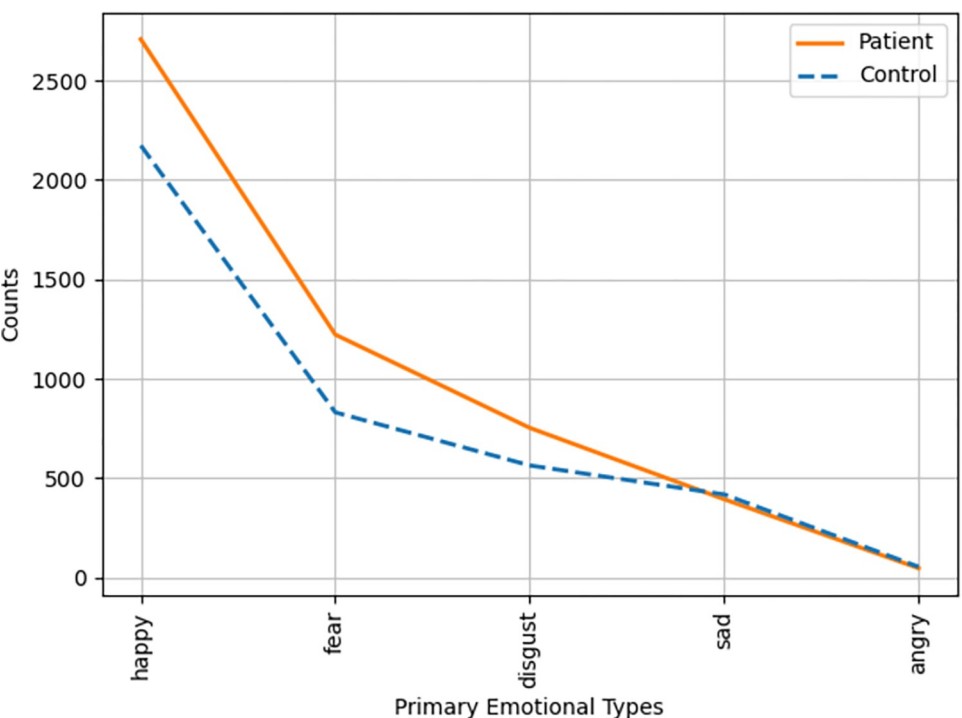

**Fig 3. Frequency of emotional words across *happy*, *fear*, *disgust*, *sad*, and *angry* between COVID-19 patients and healthy controls.**

It is worth noting that patient narratives of happiness encompass words belonging to the 'fear' and 'disgust' domains. The reason is that word identity feature was extracted from a bag-of-words model, based on the following examples:

*++HAPPY(PE)++I don't think I showed fear of death. I think death is not dreadful. We have elderly family members, so I don't think death is dreadful. We can keep a positive view about life and treat death a natural thing. I think everyone will die, and will finally reach the end of life. This is a natural process. For me, death is not dreadful. (Patient 13)*

*++HAPPY(PE)++ I never felt irritable in mood; I felt peaceful during hospital treatment. At the beginning of hospital isolation, I showed severe symptoms and event felt difficult to breathe and to eat. However, three days later I felt like eating an ox, and I could eat meals of two persons every day. I felt happy with good appetite and my immunity enhanced. I recovered very soon. (Patient 27)*

In the preceding two excerpts, Patient 13 and Patient 27 described their positive emotion toward COVID-19 infection. They demonstrated no fear toward death, and felt 'peaceful' with the virus. These two narratives of happiness encompass negative words in the domain of fear and disgust, such as 'fear', 'dreadful','die', 'irritable', 'severe symptoms', and 'difficult'. However, they are syntagmatically combined with words of negation and represent patients' positive emotions concerning their confidence in recovery.

## Discussion

Research into mental and physical health relating to the COVID-19 pandemic has applied text analysis to social media data such as Twitter, Facebook, and Reddit to forecast the possible

**Table 7. Significant word identity features between COVID-19 patients and healthy controls.**

| *Narratives of Anger* | |
|---|---|
| Features more common in Patients | |
| Words | *We* |
| Feature more common in Controls | |
| Words | *Some, news, hmm, anger, Red Cross, very, will, should* |
| **Narratives of Disgust** | |
| Features more common in Patients | |
| Words | *We, eat, I* |
| Feature more common in Controls | |
| Words | *Some, buy, Wuhan citizen, think, ah, more* |
| **Narratives of Fear** | |
| Features more common in Patients | |
| Words | *No, test, I, again, days, positive, return, nucleic acid, negative, illness, fever, infection, husband, isolation, nucleic acid test, scared, unknown, test result, nucleic acid test, result, hotel, always, nothing, reinfection, eat, airplane, confirmed infection, diagnose, night, mind, test, problems, blood, know, end, sister, call, Hubei, blood test, medicine, Blood Transfusion, now, positive result, infectious, others, return home, Nepal, blood test, Singapore, pass, cure, negative test result, inquiry, time, domestic, public health center, hotel, suspect, report, sleep, Antibody, month, worry, scream, shout, there, reinfection, indoor, airline* |
| Feature more common in Controls | |
| Words | *Uh, is, will, some, um, patient, then, all, actually, possibly, probably, lockdown, suddenly, pandemic, you, medical staff, volunteers, remember, special, one, Wuhan, city, whole, locked, goods, sight, that, particularly, more, people, patients, works, pause, vegetables, community, graduation, living condition, only, city, one month, dwellers, protective equipment, many, inside, seafood, feeling* |
| **Narratives of Happiness** | |
| Features more common in Patients | |
| Words | *I, good, doctor, report, now, should, not, anyway, nurses, nothing, mood, question, what, isolation, ill, medical staff, examine, positive for COVID-19,body, ah, also, call, eat, discharge, domestic, blood test, negative for COVID-19,cooperate, problem, days, musical instruments, music, mind, antibody, result, really, clothing, event, suppose, public health, sleep, symptoms, later, here, never, sister, all right, think, no, fear* |
| Feature more common in Controls | |
| Words | *Ah, then, Wuhan, is, some, pandemic, actually, possible, all, um, volunteers, goods, special, community, sudden, include, will, one, medical staff, pandemic hospital, city, people, organization, later, better, medical team, building, but, protective, news, students, cat, highway, patients* |
| **Narratives of Sadness** | |
| Features more common in Patients | |
| Words | *I, no, anyway, isolation, not, for sure, again, ah, my god, think, sister, worry, infectious, others, husband, then, result, return, should, sister, two, fear* |
| Feature more common in Controls | |
| Words | *Then, ah, later, Wuhan, city, possibly, he, one, um, first aid, that, life, world, patient, like, Wuhan people, one, some, car* |

emergence of emotional disorders and mental illness [23–26]. Emotional narratives of patients with confirmed COVID-19 disease have proven to be an important marker to gain a deeper understanding of emotion-related mental health issues due to COVID-19 [22]. Drawing on the text mining approach, this study aimed to examine differences in emotional status between matched groups of COVID-19 patients and healthy controls, who narrated their lived experiences of COVID-19. As part of our established procedure to assess evoked emotional discourse, we tagged narratives of happy, sad, angry, fearful, and disgusted emotions of both groups. Automated analysis of generic features, sentiment word frequency, and significant word features in the emotional narratives between COVID-19 patients and healthy controls reflected their prevailing mental health status. As for generic features of emotional narratives,

we observed that persons afflicted by COVID-19 employed significantly more words per sentence, had significantly fewer sentences per narrative, and used significantly fewer words per narrative, as compared to healthy controls. Patients demonstrated generally lower complexity of emotional language across different types of emotional narratives as opposed to healthy controls, other than anger. It is noteworthy that patients' emotional narratives of fear, happiness, and sadness were significantly shorter compared to healthy controls. Regarding different categories of emotional words, patients generally used more emotional words, in particular, more words of fear (i.e., dread and depression), disgust (i.e., criticism), and happiness (i.e., joy, trust, and like), as compared to healthy controls. This suggests that patients' emotional symptoms at the lexical level were concentrated on the emotions of fear and disgust. The analysis of the significant word features of emotional narratives showed that COVID-19 patients talked more about themselves or their family surviving COVID-19 in terms of unexpected infection, isolation, and macro-identity [22], while healthy controls mainly talked about others.

At the sentence level, patients used longer sentences to describe their emotion, revealing that their attention to topic was not as fixed as that of healthy controls, and that their emotions were unstable. This was associated with their emotional perturbation due to the COVID-19 infection. At the narrative level, healthy controls demonstrated generally higher complexity and more expressive emotional language (i.e., narratives of fear, sadness, happiness, and disgust) than COVID-19 patients, suggesting that patients' mental health disruptions had limited the linguistic range of expression of their emotional experiences. This research finding echoes the observations of prior psychiatric studies (e.g., of schizophrenia) concerning abnormalities in the production of coherent discourse due to cognitive dysfunction involving attention and thoughts [21]. A surprising finding in the present study is that narratives of anger were longer in COVID-19 patients than in healthy controls, which was in sharp contrast to the narratives of fear, happiness, sadness, and disgust. One explanation for this finding is that the long period of quarantine and relative lack of social support led to severe symptoms of worry, anxiety, depression, anger, and irritation among COVID-19 patients, given the perception of social isolation [11]. In our interviews, most patients were infected by SARS-CoV-2 passively by their colleagues, neighbors, or family members. They denied the fact that they were infected and tended to attribute the infection to other people. As a result, the salient 'angry' emotion emerged in the minds of COVID-19 patients during the first stage of infection [17]. Furthermore, when COVID-19 patients experienced social stigmatization and extreme loneliness, it is possible that their anger would evolve into an extreme emotional disorder. Overall, our results are consistent with those of published literature, which shows that COVID-19 patients are particularly vulnerable to the emotional impact of coronavirus [11–14, 16, 19–20, 22, 29].

Our findings at the lexical level support the fact that COVID-19 patients demonstrate a higher risk of psychiatric problems such as anxiety, fear, depression and stress, as compared to healthy controls [8, 9]. The emotional words used by COVID-19 patients were denser in volume than in healthy controls. In particular, the frequency of negative emotional words of fear and disgust was much higher in COVID-19 patients than in healthy controls. It is noteworthy that COVID-19 patients used more emotional words of happiness as compared to healthy controls. This is consistent with Sun et al. [17] and Deng et al.'s [22] findings regarding the dynamic path of the emotional and mental health journey of COVID-19 patients. Specifically, at the beginning of infection, the first salient emotion that emerged in patients' minds was fear, denial, and shame. After a period of isolation in hospital, COVID-19 patients gradually accepted the reality of their diagnosis and cooperated with medical staff during their treatment. In the recovery stage, patients looked forward to a favorable test result. When they finally tested negative via nucleic acid testing, their emotions changed to happiness [17]. In our interviews, the infection and the long period of quarantine led to severe symptoms of

worry, anxiety, depression, anger, and irritation among COVID-19 patients, resulting in the perception of social isolation [11, 22]. Notably, most patients were infected with COVID-19 passively by others. They felt reluctant to accept their confirmed infection and might ascribe the "tragedy" to external circumstances. Consequently, negative emotions of fear and sadness increased in COVID-19 patients at the time of infection. When patients experienced extreme loneliness and social discrimination, their negative emotions had the potential to result in the development of an overt mental health disorder. However, after patients were cured, they felt released and extremely excited. Their emotions were transformed into positive ones. This possibly accounts for the use of more emotional words of happiness in COVID-19 patients than in healthy controls. Given the dynamic psychological and emotional journey traversed by COVID-19 patients, it can well be understood why COVID-19 patients concentrated exclusively on themselves and their own well-being, while healthy controls mainly discussed others. Thus, the complex interaction of different embodied experiences, the environment, and psychological experiences interacted together to jointly mold the different emotional experiences and emotional responses in the two matched groups of participants [29].

At the general level and in terms of clinical research, generic features, emotional word frequency, and significant word identity features in the emotional narratives seen in the present study may suggest the presence of overt emotional disorders in COVID-19 patients. These linguistic features may be closely related to patient psychological trauma. In clinical management, emotional care for COVID-19 patients is likely to be equally as important as medical care to holistically manage and cure physical illness. Positive emotions play a critical role in adjustment and rehabilitation of psychological trauma associated with infection by COVID-19 [13]. In the present study, positive emotions of COVID-19 patients were associated with enthusiastic support of medical staff and the government, as well as their own expectations around their new life after recovery. This finding is consistent with that of previous studies [13, 17, 19, 22], and suggests that it is essential to reinforce positive emotions and to relieve negative emotions of patients by providing adequate and timely social support and accurate scientific information to patients [13, 22, 38–40]. Firstly, positive social interactions with health care workers, patient peers, relatives, friends, and family members are critical for regulating patient emotional responses to COVID-19 infection. Adequate emotional care should be allocated to COVID-19 patients and within their ordinary social circle. The responsibility and duty of care for emotional healing of COVID-19 patients should not be confined to their family and friends only. It is particularly necessary that health care professionals take responsibility for follow-up support of COVID-19-infected people during their recovery journey [14]. Secondly, the government should provide accurate and timeous scientific information regarding COVID-19, and establish functionally viable policies to eliminate COVID-19-related discrimination regarding employment and isolation. Policies should also be implemented to protect the privacy of COVID-19 patients. Lastly, it is necessary for the mental healthcare system to track, assess, treat, and protect the mental health status of COVID-19-infected patients [13, 40]. Specific measures may include provision of one-on-one consultations, psychological counseling sessions, interactive group support, network consultations, and telephone consultations. These psychological interventional channels should be made available to COVID-19 patients in order to share problems and enhance emotional support [40–42].

In the era of big data, emotional discourse analysis of COVID-19 patient data based on novel machine learning algorithms, and recent developments based on natural language processing techniques are promising approaches for the design of a critical surveillance system to manage pandemic-like health scenarios. To assess the emotional status of COVID-19 patients in a comprehensive manner, large-scale emotional discourse analysis based on interviews and stored social media data can be integrated into essential surveillance tools for the management

of pandemic-related mental health issues. These measures represent a novel prediction and preventive approach to effectively mitigate emotional disorders resulting from COVID-19, and also allow timely preventive interventions to be implemented for COVID-19 patients [22, 43].

## Study limitations and future research

This study has several limitations. First, we collected the narrative discourses of 34 COVID-19 patients and 24 healthy controls by convenience sampling. Our sample size was small and not entirely balanced, and our findings may thus not be generalizable to large populations. Furthermore, the interviews were conducted during two different waves of the COVID-19 pandemic within China, namely summer and autumn-winter, across 2020–2021. The dynamic change of emotional status among the two groups across the two periods may have long-term effects. A second limitation has to do with the techniques of sentiment analysis, which largely rested on the Chinese Affective Lexicon Ontology [30] and the NLTK (Natural Language Toolkit) in Python [33]. It should be noted that semantic analysis based on deep learning methods of NLP was underestimated. Third, we investigated the emotional disorders of COVID-19 patients under collective emotions as opposed to healthy controls. Individual-level correlates, such as clinical therapy, quarantine period, education level, occupation, and income, were therefore underestimated and not presented in our data.

In light of these challenges, we hope and anticipate that similar future investigations could draw on larger sample sizes of patient narratives. Enquires along this line can control the intervals of different interviews and compare emotional symptoms of COVID-19 patients across different periods. Furthermore, recent developments in NLP techniques, such as transformer-based approaches (e.g., BERT, BioBert) and deep neural network learning can be utilized to conduct sentiment analysis of COVID-19 patient narratives. These models are able to measure both the semantics of emotional words and the context that these words occur in robustly. Finally, future sentiment analysis of COVID-19 patients can consider different individual-level, socio-demographic, and pre-existing clinical correlates, given that these factors have the potential to be used as markers to predict different emotional and mental burdens among different categories of patients [3, 43, 44].

## Conclusion

The global COVID-19 pandemic is an overt threat to the emotional well-being and the mental health status of persons afflicted by COVID-19. Our text-mining-based emotional discourse analysis focused on generic features, emotional word frequency, and significant word identity features of emotional narratives in COVID-19 patients and healthy controls. Our results indicate generally lower complexity and less expressive emotional narratives in COVID-19 patients than in healthy controls. This reveals that psychological disruptions appear to inhibit the production of a detailed and coherent emotional discourse in COVID-19 patients. At the lexical level, COVID-19 patients used more words of fear, disgust, and happiness, as compared to healthy controls. Furthermore, patients narrated more words with respect to self or family while healthy controls talked mainly about others. Overall, while the emotional symptoms of COVID-19 patients at the lexical level focused on the negative emotions of fear and disgust concerning their confirmed infection and long period of treatment, their emotion was positively regulated by their recovery and adequate support from medical staff [22].

Our emotional discourse analysis may be used to distinguish the clinical status of COVID-19 patients, and can thus be used to predict the extent of emotional disorder symptoms in these patients. In clinical practice, emotional care and psychological counseling for COVID-19

patients must be an essential component of patient management, and should be considered to be an integral central element of current and future COVID-19 management strategies [11, 40]. During the present COVID-19 era, large-scale text mining of real-world data, such as interview data, social media data, and online health forum data, reveals the possibility of the design of a central surveillance system to manage emotional and mental health issues concerning COVID-19 (and possibly other future pandemic-like) infection. This system can be harnessed to inform and guide public healthcare policy and provide useful insights into developing novel therapeutics related to mental health symptoms [25].

## Author Contributions

**Conceptualization:** Yu Deng.

**Data curation:** Minjun Park, Jixue Yang, Luxue Xie, Huimin Li, Li Wang.

**Formal analysis:** Juanjuan Chen.

**Funding acquisition:** Yu Deng.

**Investigation:** Jixue Yang, Luxue Xie, Li Wang.

**Methodology:** Minjun Park, Jixue Yang, Luxue Xie, Huimin Li, Li Wang.

**Project administration:** Juanjuan Chen, Li Wang.

**Resources:** Juanjuan Chen, Li Wang.

**Supervision:** Yu Deng, Yaokai Chen.

**Visualization:** Minjun Park.

**Writing – original draft:** Yu Deng.

**Writing – review & editing:** Minjun Park, Yaokai Chen.

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
