## [Decision Letter · Decision Letter 0]

19 May 2022

PONE-D-22-06307Emotional Discourse Analysis of COVID-19 Patients and their Mental Health: An NLP-based StudyPLOS ONE

Dear Dr. Chen,

Thank you for submitting your manuscript to PLOS ONE. After careful consideration, we feel that it has merit but does not fully meet PLOS ONE’s publication criteria as it currently stands. Therefore, we invite you to submit a revised version of the manuscript that addresses the points raised during the review process.

We look forward to receiving your revised manuscript.

Kind regards,

Giuseppe Carrà, PhD

Academic Editor

PLOS ONE

Journal Requirements:

Reviewers' comments:

Reviewer's Responses to Questions

**Comments to the Author**

1. Is the manuscript technically sound, and do the data support the conclusions?

Reviewer #1: Partly

Reviewer #2: Partly

2. Has the statistical analysis been performed appropriately and rigorously? 

Reviewer #1: Yes

Reviewer #2: No

3. Have the authors made all data underlying the findings in their manuscript fully available?

Reviewer #1: No

Reviewer #2: No

4. Is the manuscript presented in an intelligible fashion and written in standard English?

Reviewer #1: Yes

Reviewer #2: No

5. Review Comments to the Author

Reviewer #1: In the manuscript entitled “Emotional Discourse Analysis of COVID-19 Patients and their Mental Health: An NLP based Study”, the authors aimed to study potential differences in the emotional status between people who reported COVID-19 infection and healthy controls, benefiting from novel approaches based on natural language processing methods.

The topic is very interesting and the specific methodological approach appears promising in the mental health field, possibly contributing to explore emotional status and the potential link with a differential mental health burden.

Nevertheless, some changes are required to improve the manuscript quality:

• In the Introduction section, the study justification should be clearly presented, by focusing on the need to explore mental health burden or at least proxies of it ( https://doi.org/10.1016/S0140-6736(21)02143-7 )

Similarly, in the Abstract, at least one sentence is needed to describe the study justification.

• In the Materials and Methods section, the authors should better clarify the nature of the study (e.g., cross-sectional design). When were narratives collected for people who experienced COVID-19? The authors may wish to take this into account also in the Discussion section.

• Further details should be added in terms of sampling strategies and potentially sample size calculations with additional comments in the Limitations section, where appropriate.

• Similarly, NLP should be described more in detail, thus providing the reader with additional information regarding the methodological approach for features computation. For example, the authors should report whether they benefited from a specific software (and related packages) and they should add a more in-depth description of chosen techniques. In addition, they should clarify whether they combined qualitative and quantitative approaches.

• In the current version of the manuscript, the study design paragraph includes some details that are specific of the chosen methodological approach. Therefore, it would be useful to add a reference to the paragraph in which the authors describe the current approach in detail.

• More importantly, a subparagraph Measures should be added to the Materials and Methods section.

• Page 3 lines 146-149: It seems more like a result rather than a methodological description. Please check.

• Lines 162-180: How were interview's domains defined? Please clarify.

• The rationale of choosing two data collection periods (i.e., Summer and Autumn-Winter) should be clarified. Did the authors check data for differences across the two periods? I suggest discussing this issue further.

• In Table 2, what do the initials in brackets mean? Please add relevant footnotes.

• Some details mentioned at the beginning of the Results section need to be disclosed in the Materials and Methods section. Please revise the logical flow of the Methods section to include all relevant methodological details in order to let the reader fully understand the analyses.

• Moreover, the techniques to implement sentiment analyses are not clearly presented in the Methods. Please revise related sentences, also referring to commonly used approaches such as the Valence Aware Dictionary for sEntiment Reasoning (VADER) and the Bidirectional Encoder Representations from Transformers (BERT) ( https://doi.org/10.1192/j.eurpsy.2021.3 )

• There is some ambiguity between the two paragraphs of the Results section. For example lines 274-284 seem to fit best the next heading (Sentiment Frequency of emotional words and word identity). Please clarify.

I suggest providing a clearer definition of the measures in the Materials and Methods section (see my previous comment about an additional “Measures” paragraph) and then following an unequivocal logical flow in the Results section according to what is described in the Methods. Moreover, types of emotional words (see Figure 4 footnotes) should be described in the Materials and Methods section.

• When interpreting Table 4, the authors focused on some emotions according to statistically significance (0.05 threshold). However, table 4 seems to show an additional perspective about potential emotional differences between patients and controls. In particular, although some estimates were not statistically significant, it shows that, as a whole, controls were likely to express different emotions through a higher number of words as compared with patients, except for anger. Although these results should be interpreted with caution, this perspective should be taken into account. I suggest the authors to revise relevant sentences in the Results section and add a comment on that in the Discussion section. Do you think that this might be related to the time interval between COVID-19 infection and the interview? In other words, can this be considered a short- or also a long-term effect? What are potential explanations? Please add a comment in the Discussion section, also considering available information from collected data.

• Furthermore, the authors should clarify whether they collected meta data such as information on therapy/recovery, but also gender, age, etc… or they should acknowledge the lack of key information as a study limitation. Individual-level correlates may play a role that needs to be taken into account for future research. Indeed, based on available evidence, different individual-level, socio-demographic and pre-existing clinical correlates are likely to be associated with an increased mental health burden and should be taken into account ( https://doi.org/10.1016/S0140-6736(20)30460-8 ; https://doi.org/10.1016/j.neubiorev.2021.10.010 ; https://doi.org/10.1016/S0140-6736(21)02143-7 )

• Lines 286-287: “We focused exclusively on examining lexical features in the emotional narratives related to the emotionality.” Please clarify. In addition, did the authors mean frequency distribution or rather sentiment polarity?

• Qualitative results are not adequately justified through representative excerpts from the data. Therefore, some excerpts from the qualitative analysis should be provided and discussed in order to let the reader better understand the results, along with those provided next to Table 2.

• The Discussion section appear to jump to conclusions. These need to be clarified and justified. For example lines 351-352; 356-358 (Sentiment or content analysis?); 362-364; 371-373. Please give additional explanations and consider providing adequate references. As a result, the first paragraph of the Discussion needs to be extensively revised.

• How was happiness combined with emotional words belonging to fear and disgust domains? In other words, happiness is somehow opposite to fear and disgust. How did the authors interpret these findings? (Consistently, please report the name of the emotion.)

• Furthermore, based on available evidence, the authors should add a comment in the Discussion section on the use of novel machine learning algorithms and relevant methods grounded on natural language processing techniques as a promising approach for the design of a critical surveillance system to manage pandemic-like scenarios. These considerations should contribute to emphasize the importance of an emotional discourse analysis that may possibly benefit also from social media integration for timely preventive interventions ( https://doi.org/10.1192/j.eurpsy.2021.3 )

• I suggest revising the Conclusion paragraph, by making the first part more compact in order to let the reader better understand the take home message.

Reviewer #2: In the study presented in this article, the authors used semi-structured interviews and affirm to use natural language processing to analyze the characteristics of lived experience narratives between patients with COVID-19 and a group of healthy patients related to five basic emotions. The aim of the work was to identify differences in emotional state between the two groups of participants. The results obtained indicate generally higher complexity and more expressive emotional language in the healthy patient group than in the COVID-19 patients.

The work has a rather relevant objective but, in my opinion, it fails to be convincing with respect to the results obtained, this due to several weaknesses and naive solutions used in the interpretation of the results.

- First of all, semi-structured telephone interviews were used in the study. It is not discussed sufficiently how it is ensured that the interview modality and the questions asked do not in any way imply bias with respect to the answers provided. Has it been thought to also consider other content generated by these users independently of the telephone interview? There are works that have tried to assess the psychological vulnerability of users with respect to this aspect, and other works more similar to the proposed work, which were not mentioned in the work, for example:

* Low, Daniel M., et al. "Natural language processing reveals vulnerable mental health support groups and heightened health anxiety on reddit during covid-19: Observational study." Journal of medical Internet research 22.10 (2020): e22635.

* Patel, Rashmi, et al. "Analysis of mental and physical disorders associated with COVID-19 in online health forums: a natural language processing study." BMJ open 11.11 (2021): e056601.

* Viviani, Marco, et al. "Assessing vulnerability to psychological distress during the COVID-19 pandemic through the analysis of microblogging content." Future Generation Computer Systems 125 (2021): 446-459.

It is true that the authors take into consideration specific groups of users affected and not affected by COVID-19, however there are interesting aspects of the previous works that could be taken into consideration, the lack of which constitutes another weak point of the presented work.

- In fact, the authors claim to use "the natural language processing method". Regardless of the fact that the sentence itself is incorrect, as NLP includes a series of techniques for natural language processing (and it is not a "method"), the fact remains that the text analysis method used by the authors is rather simple and may in any case refer to techniques of "text mining" rather than "natural language processing". In fact, no semantic or natural language understanding analysis is carried out, as frequentist analyzes of the appearance of particular textual characteristics within the interviews are essentially taken into consideration.

- In evaluating users' emotions, the authors therefore follow an approach based essentially on the lexicon, without making any reference to recent developments in the NLP field which take into account both the semantics of words and the context in which these words appear (BERT, BioBERT , transformer-based approaches, use of deep neural networks).

Ultimately I can say that the work from the point of view of language analysis with respect to the problem in question is not particularly innovative compared to previous works of literature, and, in any case it refers to concepts that are ultimately not really used within the work (for example NLP).

I also believe that on a rather limited group of patients it is difficult to draw conclusions that are statistically relevant, even this aspect should have been discussed more carefully.

Finally, I must say that the level of English should also be improved, because very often there are sentences that are not understandable or at least not in standard English.

6. PLOS authors have the option to publish the peer review history of their article (what does this mean?). If published, this will include your full peer review and any attached files.

Reviewer #1: No

Reviewer #2: No

---

## [Author Response · Author response to Decision Letter 0]

23 Aug 2022

Please see the "Response to Reviewer comments" in the attached file

---

## [Editor Report · Decision Letter 1]

25 Aug 2022

Emotional Discourse Analysis of COVID-19 Patients and their Mental Health: A Text Mining Study

PONE-D-22-06307R1

Dear Dr. Chen,

We’re pleased to inform you that your manuscript has been judged scientifically suitable for publication and will be formally accepted for publication once it meets all outstanding technical requirements.

Kind regards,

Giuseppe Carrà, PhD

Academic Editor

PLOS ONE
---

## [Editor Report · Acceptance letter]

8 Sep 2022

PONE-D-22-06307R1 

Emotional Discourse Analysis of COVID-19 Patients and their Mental Health:
A Text Mining Study 

Dear Dr. Chen:

I'm pleased to inform you that your manuscript has been deemed suitable for publication in PLOS ONE. Congratulations! Your manuscript is now with our production department. 

Kind regards, 

on behalf of

Dr. Giuseppe Carrà 

Academic Editor

PLOS ONE